# Membrane Distillation Crystallizer Applied for Separation of NaCl Solutions Contaminated with Oil

**DOI:** 10.3390/membranes13010035

**Published:** 2022-12-28

**Authors:** Marek Gryta

**Affiliations:** Faculty of Chemical Technology and Engineering, West Pomeranian University of Technology in Szczecin, Piastów Ave. 42, 71-065 Szczecin, Poland; marek.gryta@zut.edu.pl

**Keywords:** membrane distillation crystallizer, scaling, brine, wettability, oil fouling

## Abstract

In the present study, the membrane crystallizer was used to separate a saturated NaCl solution contaminated with an oil emulsion. The crystallizer was connected via a mesh separator with a feed tank in which capillary submerged modules were assembled. The effect of scaling and oil sorption on the wetting of polypropylene (PP) membranes has been investigated during the long-term studies. It has been found that cooling the solution in the crystallizer by 15 K below the feed temperature resulted in intensive NaCl crystallization in the zone below the mesh separator. A result, the salt crystallization on the membrane surface was eliminated. Contamination of saturated brines with oil in the concentration exceeding 100 mg/L caused the oil penetration into the membrane pores. The application of a PP net assembled on the capillary membranes surface reduced the intensity of wetting phenomenon caused by scaling and the oil sorption, which provides a stable membrane module performance during 1300 h test.

## 1. Introduction

In the membrane distillation (MD), in contrast to reverse osmosis, the osmotic pressure does not reduce the driving force [1]. For this reason, the MD process is often proposed for the concentration of salt solutions [1,2,3,4]. The preparation of highly concentrated brines allows to obtain a high degree of water recovery in the desalination process. This advantage means that the MD process can be used to implement the concept of “Zero Liquid Discharge (ZLD)” [5,6]. However, the main challenge to implementing this technology is scaling, which hinders achieving the long-term durability of MD membranes.

The basis of the MD process is the preservation of non-wetted pores in the membranes [7,8]. While concentrated unsaturated salt solutions do not wet the hydrophobic membranes [9,10], wetting of the pores can occur when the salt crystallizes on the membrane surface [11,12,13]. For this reason, the membranes were wetted faster during a vacuum MD variant than in a process carried out using a direct contact MD configuration [14]. As a result of pores wetting, the salt crystallizes not only on the surface of the membranes, but also inside the wall, which damages the MD membranes [5,12,15,16]. For this reason, not only the limitation, but essentially the elimination of scaling enables the application of MD technology on an industrial scale. The membrane fouling can also be caused by other substances present in the brines, hence**,** for actuals brines (e.g., RO retentate) an additional feed pre-treatment is required [17,18,19].

Foulants removal by brine pre-treatment does not eliminate the scaling issue since the salt solubility is often exceeded during the feed concentration in the ZLD process [5,20]. The separation possibility of the saturated solutions can be achieved by reducing the salt concentration in the feed using a crystallizer attached to the MD installation (membrane distillation crystallizer—MDC) [15,16,21]. In the MDC system, the brine concentrated in the MD installation flows into the crystallizer, where the supersaturated solution becomes saturated due to crystallization. As a result of salt concentration in the MD modules, the nucleating agents are formed, which accelerates the salt crystallization in a cooled crystallizer [21,22]. The mother liquor is heated (thus stays undersaturated) and is returned to the MD installation after crystallization. This arrangement avoids scaling if the solubility of the salt changes significantly with the temperature of a solution, as in the case of KCl or Na_2_SO_4_ [21,22,23]. Unfortunately, the temperature has little effect on the NaCl solubility, which makes it difficult to eliminate scaling during the separation of its solutions in an MDC installation consisting of a crystallizer connected with MD module [20,24,25].

NaCl crystallizes at a small supersaturation ratio (S), and its limit for spontaneous crystal formation (primary nucleation) is S = 1.0025 [20]. Therefore, in order to maintain the continuous operation of MDC the careful adjustment of the process parameters is required. Maintaining the appropriate parameters is rather difficult, therefore, in many studies, the MDC system was tested in a batch arrangement [15,22]. In this case, the concentration of the feed was carried out to the supersaturation state and the MD process was interrupted after the crystals formation in the feed tank. Then, in order to eliminate scaling, the saturated feed was removed followed by rinsing the MD modules with water [26,27], which generated a waste brine. More suitable solution is the semi-batch system in which a continuous flow of unsaturated feed through the MD modules can be obtained by using two alternating tanks (one—feed concentration, and second—feed cooling and crystallization) [25,28]. However, during the period when the feed is concentrated to the saturation state, no salt crystallization occurs, which reduces the efficiency of the proposed system. Hence, a continuously operating MDC has a much greater efficiency. In this case, the scaling was limited by separating the feed concentration zone from the crystallization zone using the flow reactors [21]. Similar beneficial effects were achieved using a multi-stage cascade system [29]. The use of two additional tanks in addition to the crystallizer allows to obtain a zone of dilution and elimination of the nucleation seeds in a feed flowing to the MD modules. The disadvantage of using multi-tank cascade systems is not only the increase in the volume of the installation, but also the increase in heat losses resulting from increased size of such system. In this study, a novel two-chamber solution for the MDC installation was developed. A separator consisting of several meshes was installed on the outflow from the crystallizer, which allowed to eliminate the flow of salt crystals to the MD chamber.

In addition to the inorganic scaling, a major difficulty in the implementation of the MD process is the presence of organic compounds. When such compounds contaminate the feed, their concentration in the brines progressively becomes higher during MD process operation. As a result, the surface tension of the feed decreases, which may cause wetting of the hydrophobic membranes [11]. The presence of oils and surfactants in the feed is particularly dangerous for the durability of MD membranes. They facilitate the penetration of the feed into the hydrophobic membranes, which leads to their rapid wetting [7,11,30]. This problem was solved by using the membranes with modified anti-wetting surface properties [31]. However, despite the many advantages of the new types of membranes, there have been no reports of their use in the pilot plant MD installations. Nowadays, these installations are equipped with the modules having the membranes made of polypropylene (PP), polyvinylidene fluoride (PVDF), polytetrafluoroethylene (PTFE) and polyethylene (PE) [20,26,32]. For this reason, developing a novel membrane research, especially long-term, is necessary for MD process industrial implementation. In this work, the protective effects of net wounded on the capillary membrane was presented.

The desalinated water can also contain petroleum pollutants, which accelerate the wetting of membranes at high salt concentrations in the MDC installation [19]. A previous work has shown that the presence of small amounts of oil (up to 100 mg/L) forming an emulsion with micrometer droplets does not wet the PP membranes [11,33]. The hydrophilic groups, such as carboxyl groups, are formed on the polymer surface during the MD process, which gives the PP membrane some oleophobic properties [33]. Strongly hydrophobic membranes do not possess such properties, hence a much higher wetting of PTFE membranes was noted with a lower oil content in the treated feed [19].

The membrane wetting can proceed very slowly, hence in this work the polypropylene membranes were tested in the MDC installation for over 1000 h. The concentrated NaCl solutions contaminated with oil emulsion were used as a feed. The process was carried out in a two-tank installation connected via a mesh separator. The capillary submerged modules were used to separate the feed. Additionally, the anti-wetting/scaling effects of protective polypropylene net assembled on the surface of the capillary membranes were studied.

## 2. Materials and Methods

The separation of concentrated NaCl solution by MDC were carried out in the installation presented in Figure 1. It consisted of two tanks (4 L) connected by an overflow (1 inch diameter tube). The tank (7) serves as a stirred tank crystallizer and it was placed on the magnetic stirrer RCT Basic (IKA-Werke GmbH, Staufen, Germany—700 W, 800 rpm). The stirrer had a temperature sensor which allowed to regulate the brine temperature. Under the overflow, four layers of polyamide meshes with 120 μm mesh size were installed, filling the entire cross-section of the tank. The crystallizer was cooled with a polyethylene spiral in which tap water flowed. The temperature (T_F_) in the feed tank (1) was controlled by an electrical heater (1000 W) integrated with the EDIG temperature controller (Nüga Company, Georgensgmünd, Germany). The feed was pumped by a peristaltic pump (100 mL/min) to the crystallizer, in which its retention time was 40 min. The MD tests were run continuously. Once a day, the distilled conductivity and the permeate flux were determined.

In the present study, the Accurel PP V8/2 polypropylene capillary membranes manufactured by Membrana GmbH (Wuppertal, Germany) were tested. These membranes are fabricated by thermally induced phase separation (TIPS) and exhibited a partial oleophobic performance [33]. The capillaries inner diameter and the wall thickness were equal to 5.5 mm and 1.5 mm, respectively. The membranes were characterized by sponge structure with porosity of about 70% and the average pore size amounted to 0.22 μm (manufacture’s specification).

These membranes are also produced with a net assembled on the surface (Figure 2). The net is made of fine polypropylene fibers (40 µm). The fiber bundles were wound very tight on the capillary. A part of fibers press into the membrane surface and deform it, creating characteristic grooves (Figure 2b). In the work, this membrane was marked with the symbol V8N, and the membrane without a net was marked as V8. Several similar modules which were made by gluing the ends of a single membrane into glass tubes (diameter 5 mm) were used for the MD tests. The capillary active length was 25 cm, and the outer area was 67 cm^2^.

The scanning electron microscope (SU8000, Hitachi High Technologies Co., Tokyo, Japan) was used to investigate the membranes morphology. The membrane samples were sputter coated with chromium. In order to assess the degree of pores wetting, changes of the distillate electrical conductivity were systematically measured using the 6P Ultrameter apparatus (Myron L Company, Carlsbad, CA, USA). The oil concentration was determined by an oil analyzer OCMA 500 (Horiba, Kyoto, Japan). This apparatus performs an infrared analysis after an automatic extraction of the oil from the water sample with the solvent S316 (Horiba).

## 3. Results and Discussion

### 3.1. MD of NaCl Solutions Contaminated by Oil

In order to compare the effects of using a two-chamber MDC installation with a mesh separator, in the first series of measurements, the separation of the NaCl solution was carried out in the installation without a crystallizer. In this case, using an M1 module made of Accurel PP V8/2 membrane (V8), the feed was concentrated to the supersaturated state in its tank. The research started with a NaCl solution containing 300 g/L of salt. The addition of the oil emulsion (80 mg/L) to the feed reduced the permeate flux from 1.3 to 1.1 L/m^2^h, which was on the following day equal to 1.06 L/m^2^h (Figure 3). A similar decrease in the efficiency for NaCl solutions containing 100 mg/L of oil was found in [11]. This work showed that the oil was adsorbed on the PP surface, which blocks the feed flow to the evaporation surface and, as a result, reduces the permeate flux.

After 80 h of the MD process run, salt crystallization was visually observed both on the feed tank walls and the V8 membrane surface. As a result, the permeate flux decreased from 1.06 to 0.96 L/m^2^h, and after the next three days further decreased to 0.4 L/m^2^h was noted (Figure 3, from 150 h). During this period, the conductivity of the distillate also increased, which indicated that the changes in flux resulted not only from scaling but also from a partial wetting of the membrane. The addition of oil to the NaCl solution reduces the surface tension of the feed [11,19], which could facilitate wetting of the pores. As the thickness of a salt layer on the membrane surface was increased, the module efficiency declined. The tests were completed with a flux of 0.05 L/m^2^h and the conductivity of the distillate increased to 246 μS/cm. The increase in conductivity accelerated significantly after 200 h MD, most likely due to scaling which has been repeatedly shown to facilitate wetting of the MD membranes [14].

The SEM studies confirmed the intensive V8 membrane scaling (Figure 4). The salt crystallized on the membrane surface (Figure 4c) and inside its wall (Figure 4d). Crystals in the pores not only block the vapor flow, but also cause a mechanical destruction of the pores. For this reason, scaling causes a damage of the MD membranes apart from wetting hydrophobic membranes [12,13]. SEM-EDX analysis of the Na and Cl content in the cross-section of the membrane showed that the salt penetrated more than 500 µm into the wall (Figure 4e,f). Such a result, obtained after less than 400 h of MD process, suggests that the implementation of a continuous salt crystallization in a single-chamber MDC system will be impossible, even using thick-walled membranes.

### 3.2. Two-Chamber MDC Installation

In the second stage of this study, the MDC installation shown in Figure 1 was operated, but with only one module (M2—membrane V8). The temperature of the NaCl solution (300 g/L) in the feed tank (T_F_) and crystallizer (T_C_) was equal to 333 K and 318 K, respectively. In this case, the temperature difference ∆T = T_F_ − T_C_ was equal to 15 K. The volume of the solution was kept constant in the installation and its losses were replenished by continuous dosing the feed tank with NaCl solution (200 g/L). After two days of MDC operation, the salt crystals appeared in the crystallizer and the permeate flux stabilized at a level of 1.2 L/m^2^h (Figure 5).

Contrary to expectations, the deposition of salt on the surfaces of both cooling coil and the mesh separator (Figure 1) was slight, although their surfaces could promote the heterogeneous nucleation. A thin layer of fine crystals was formed mainly on the surface of the meshes which, however, did not block the flow of the mother liquor. As a result of the stirrer operation, most of the salt crystals formed a suspension in the zone under the meshes, and some of the crystals sedimented at the tank bottom. The liquid flowing through the mesh separator was clear without the crystals and the mother liquid discharged from the crystallizer flowed smoothly over the overflow into the feed tank. In the work [20], for similar values of the permeate flux and brine temperature, most of the particles present in a suspension, apart from single crystals with cubic habit, were agglomerated with sizes over 200 µm. The mesh openings in the assembled meshes had a size of 120 µm, which enables such crystals to be retained. As demonstrated in [27,34], the use of a crystal retaining filter in front of the MD module can significantly reduce the occurrence of scaling. The feed tank was heated, hence**,** the solution temperature (T_C_) increased from 318 to 333 K (T_F_). This temperature changed the feed state from saturated to unsaturated, and as a result, no salt crystallization was observed.

After 120 h of MDC experiments, the oil emulsion was added to the NaCl solution, resulting in an oil concentration of 82 mg/L. As a result, the permeate flux slightly decreased and was stabilized at the value of 0.95 L/m^2^h after two days and amounted to 0.9 L/m^2^h after almost 300 h of the process (Figure 5). The distillate conductivity increased to 32 μS/cm after this period. It should be noted that the conductivity value was four fold lower than that observed after 300 h of the operation run using the installation without a crystallizer (Figure 3). With a similar oil content, a decrease in the permeate flux was also significantly lower. This indicates that the main reason for the membrane failure in the module M1 was the scaling but not the presence of the oil emulsion. The obtained results showed that a two-chamber MDC solution used enables the implementation of a continuous MD process of the brines with a simultaneous salt crystallization.

The SEM examination of the membrane samples collected from the M2 module confirmed that the internal scaling was eliminated. The surface of the membrane was mostly clean, with only a few NaCl crystals spotted in some places (Figure 6). In the case of MDC, the effect of concentration polarization may cause the salt solubility at the feed/membrane interface to be exceeded even when a feed flow turbulence was increased [14,20,35].

In the MDC installation, the scaling intensity depends on the permeate flux (concentration polarization) and the temperature in the crystallizer, which determines the concentration of the brine flowing into the MD chamber [34]. For this reason, a less intensive scaling was found after reducing the feed temperature and increasing the cooling of the crystallizer [20,34]. In the previous series (Figure 5), the feed temperature was 333 K, whereas the temperature of the crystallizer was 318 K, i.e., it was lower by ∆T = 15 K. In order to limit the scaling, in subsequent tests, the feed temperature was reduced to T_F_ = 328 K and the crystallizer cooling was more intensive to reach the temperature T_C_ = 310 K (∆T = 18 K). The installation was filled with NaCl solution, saturated at 310 K. The initial oil content in the feed was 85 mg/L. For these MDC operating conditions, significant amounts of salt crystals were formed in the crystallizer when the process was carried out for one day. In this series of tests, in addition to the M3 (V8) module, the M4 module with the V8N membrane covered with a net shown in Figure 2 was also installed in the MD chamber. Salt crystallization on the membranes surface causes the wetting of the surface pores [12,13]. The net assembled on the membrane surface could provide a substrate for crystal formation and limit their contact with the surface pores.

For the applied MDC parameters (T_F_ = 328 K and T_C_ = 310 K), the permeate flux was equal to 0.75 L/m^2^h (M3) and 0.59 L/m^2^h (M4). Thus, the decrease of T_F_ from 333 to 328 K resulted in over 15% permeate flux reduction. After 300 h of the process (Figure 7)**,** the above mentioned values decreased to 0.65 L/m^2^h (M3) and 0.44 L/m^2^h (M4). The net on the membrane surface increases both the temperature and concentration polarization, hence the efficiency was lower for the M4 module. The obtained results showed that there were smaller decreases in the permeate flux for both modules compared to a decrease observed for the higher temperature T_F_ = 333 K (Figure 5). The distillate conductivity was also several times lower (only up to 6 μS/cm), which indicates that the scaling did not cause wetting of the pores. The obtained results confirmed that the reduction of the process temperature had a positive effect on the separation process of the saturated NaCl solution in MDC installation [34].

The SEM studies also confirmed that the scaling can be eliminated by lowering the process T_F_ temperature (from 333 to 328 K). No NaCl crystals, previously found in the M2 module (Figure 6), were observed on the surface of the V8 membrane (Figure 8a). The presence of a net on the membrane surface increases the polarization phenomena. Thus, the conditions for salt crystallization inside the net and on the membrane surface can be created due to this phenomenon. Moreover, no salt crystals were found on the V8N membrane surface after the removal of net (Figure 8b).

### 3.3. Long-Term MDC Studies

The brines contaminated up to 100 mg/L of oil separated by the MD process conducted for several thousand hours showed a good wetting resistance of capillary PP membranes [11,33]. However, some unfavorable changes were noticed after several hundred hours of MD, when the concentration of NaCl in the feed was 300 g/L [11]. Similar and higher salt concentrations occur in the MDC installation, hence the long-term investigations were carried out in the last stage of this study to determine the stability of the process. A feed temperature was kept as in the previous series (T_F_ = 328 K), but a crystallizer temperature was increased from 310 to 318 K, which gave ∆T = T_F_ − T_C_ = 10 K. Increasing the crystallizer temperature allows to reduce energy losses in MDC [25]. On the other hand, this change increases the concentration of the salt solution returning from the crystallizer to the MD chamber, which can increase the negative effect of concentration polarization [34]. For this reason, 1300 h research was carried out to determine the impact of the new MDC operating conditions on the membrane wettability, the results of which are presented in Figure 9 and Figure 10.

The MDC installation was filled with NaCl solution saturated at 318 K and the tests were started without the addition of oil. The M5 (V8) and M6 (V8N) modules were placed in the MD chamber. A crystal suspension was formed in the crystallizer after the day of the process. The initial permeate flux was close to the value obtained in the previous series of tests (Figure 7). However, in the following days, the efficiency of modules was systematically decreasing (Figure 9) and after 200 h of the process it stabilized at the level of 0.34 L/m^2^h (M5) and 0.2 L/m^2^h (M6). This result indicates that increasing the crystallization temperature from 310 to 318 K could facilitates the surface nucleation and the resulting salt deposits blocked the membrane surface. Scaling, confirmed in the further part of the study by SEM observations, resulted from an increase in the concentration polarization [34]. During this period, the distillate conductivity was below 10 μS/cm (Figure 9, up to 300 h) thus the membranes were not wetted. A net assembled on the surface of the V8N membranes additionally increased the polarization phenomena, which could result in greater salt crystallization on the membrane surface. Therefore, the initial permeate flux obtained for the V8N membranes was about 30% lower than that obtained for the V8 membranes.

After 300 h of the MD process run, the oil (78 mg/L) was added to the feed and the efficiency of the M5 (V8) module started to decrease again (Figure 9). As a result, after 700 h MD, the permeate flux obtained for M5 was the same as for the M6 module (V8N). At the same time, the conductivity of the distillate increased (up to 50 μS/cm), which indicated that the partial wetting of the V8 membranes could be a reason of a decrease in the M5 efficiency. In contrast, for M6 module with the V8N membrane, the permeate flux at this time was stable. This indicates that the assembled net limited the negative impact of the feed, especially the effects of salt crystallization. This is also confirmed by the fact that the conductivity of the distillate for M6 module only increased to 18 μS/cm.

Scaling hindered the feed flow, especially when the salt crystals were formed directly on the membrane surface in the M5 module. To confirm this thesis, the concentrated brine was replaced with an unsaturated NaCl solution (100 g/L), which should dissolve the deposits on the membrane surface. Indeed, the permeate flux increased to a level above 1.2 L/m^2^h and increased over the next two days, which indicated the purification of the membranes in the modules (Figure 9, point D). At the same time, the conductivity of the distillate decreased below 5 μS/cm, which is attributed to the fact that only a few pores in the V8 membrane were wetted. It should be noted, however, that the salt deposit formed on the membranes was very thin as it was not visually noticeable. This was a different result than that obtained in the previous tests (without MDC) for the membrane placed in the M1 module, on the surface of which visible large NaCl crystals were formed.

In the following days, the MDC installation was refilled with a saturated NaCl solution with oil (64 mg/L). The initial flux values for the washed membranes were slightly lower than those obtained for the pure NaCl solution (Figure 10) and amounted to 0.38 L/m^2^h (M5) and 0.3 L/m^2^h (M6). During 100 h of process, the permeate flux of both modules was stable although it decreased to 0.31 L/m^2^h (M5) and 0.28 L/m^2^h (M6) when the oil content was increased to 94 mg/L in the feed (Figure 10, point 2). For the next 100 h of process run, the permeate flux stabilized at the level of 0.22 L/m^2^h (M5) and 0.23 L/m^2^h (M6) (Figure 10, 1100 h). After adding another portion of the oil emulsion (point 3–124 mg/L), the efficiency of the M6 did not decrease, in contrast to the M5 module with the V8 membrane, for which the permeate flux decreased to 0.03 L/m^2^h during 90 h period. At the same time, the conductivity of the distillate increased to 2300 μS/cm, which indicated a significant wetting of the V8 membrane in the M5 module. A similar wetting of the membranes for the feed containing oil over 100 mg/L during the brines separation was reported by Kim at al. [19]. Increasing the oil emulsion content in the feed facilitates the coalescence of oil droplets on the membrane surface, the intensity of which increases with the salt concentration [11]. The large droplets penetrate more easily into the pores of the PP membrane, to which access was limited by the net assembled on the surface of V8N membrane, hence the efficiency of the M6 module did not decrease.

The SEM studies showed that a decrease in the permeate flux presented in Figure 10 was resulted not only from fouling but also from the changes in the membrane structure. After completion of the tests, the surface of the V8 membrane in the M5 module (Figure 11a) was much less porous than the V8N membrane (Figure 11b). The SEM-EDX studies confirmed that on the V8 membrane surface was deposited NaCl (Figure 11c,d), however its amount was much less than those formed on the membrane surface in the M1 module (Figure 4). This result shows that the use of a two-chamber MDC installation allowed to significantly reduce the membranes scaling. The salt deposits were practically not found on the surface of the V8N membrane (Figure 11b). In this case, the salt crystallized on the surface of the net fibers (Figure 12a). The nets, similar to the membrane, were made of PP, hence the analysis of the content of C (Figure 12b(C K)) allowed the determination of the places covered with minerals (darker places). Their localization was consistent with the increase in the intensity of the colors depicting the presence of Cl and Na. The performed studies have shown that the sediments also contained Si.

Although a salt deposit was found on the V8 membranes surface, a small amount of it did not completely cover the surface (Figure 11c,d), therefore the deposit should not cause such a significant drop in the permeate flux as was indicated (Figure 10). The SEM observations of the membrane cross-sections showed that the reason for the flux decline can be associated with the changes in the pore structure inside the membrane wall (Figure 13). The structure changes occurred in the outer wall layer to a depth of up to 400 µm, which manifested itself in the form of a “dark ring” (Figure 13a). Inside this layer, small amounts of NaCl crystals were found, mostly near the external edge (Figure 13b). The presence of salt indicates that this pore layer was wet, but there was no salt in the deeper layers of the “dark ring”. This was confirmed by the results of the SEM-EDX analysis, showing the presence of salt only near the membrane surface (Figure 13d). Probably, the observed effect was due to the penetration of oil into the wall, which changed the structure of the membrane matrix. As a result, the pores became more closed (Figure 13e). The pores in the wall below the “dark ring” were definitely more open (Figure 13f), such as those observed in the new membrane.

Regarding the V8N membrane, no penetration of the feed into the wall was found, the pores throughout its cross-section remained unchanged and were similar to those shown in Figure 13f. This indicated that the PP mesh assembled on the membrane surface prevented oil from penetrating into the pores.

## 4. Conclusions

The conducted research confirmed that the application of a crystallizer attached to the MD installation ensures the effective separation of saturated NaCl solutions. It is important to provide the conditions of the undersaturation state in the solution contacting the surface of membranes. Such conditions were achieved in the presented two-chamber MDC installation with a mesh separator through which flowed the mother liquor returning from the crystallizer to the MD chamber.

A temperature of the solution in the crystallizer affects the intensity of scaling phenomenon. When it was only about 10 K lower than the feed temperature in the MD chamber (∆T = T_F_ − T_C_), salt crystallized on the membranes surface, which was definitely limited for ∆T = 15 K and can be completely eliminated for the temperature difference equal to 18 K.

The presented study has shown that when the saturated NaCl solutions become contaminated with an oil emulsion, the oil can penetrate the pores of PP membranes resulting in the changes of the membrane matrix. As a result, the PP membrane lost its permeability. For this reason, the oil content in the mother liquor should be definitely below 100 mg/L.

Although the assembling of PP net on the capillary membranes surface reduced the permeate flux by 30%, the membranes pores can be protected against scaling and oil sorption. Consequently, a stable module performance with the V8N membrane was obtained also under unfavorable process conditions, in which the membrane without protective mesh (V8) lost their permeability during the 1300 h process.

## Figures and Tables

**Figure 1 membranes-13-00035-f001:**
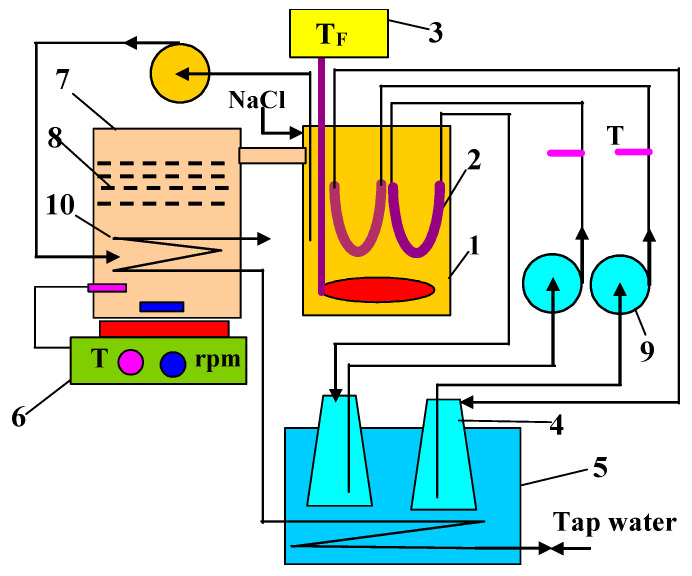
Experimental MDC set-up. 1—feed tank, 2—submerged MD module, 3—feed temperature controller, 4—distillate tank, 5—cooling bath, 6—magnetic stirrer with heating plate, 7—crystallizer, 8—poliamide meshes, 9—peristaltic pump, 10—cooling coil T—thermometer, NaCl—make-up solution (200 g/L).

**Figure 2 membranes-13-00035-f002:**
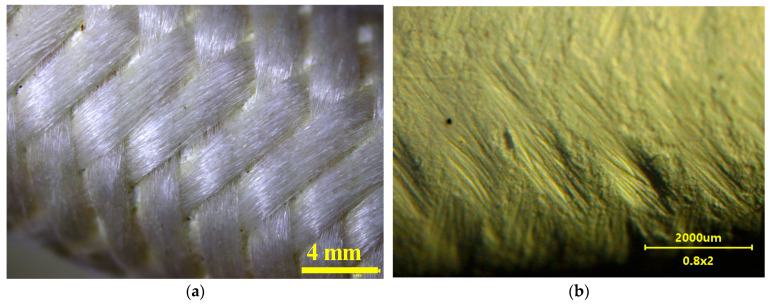
Membrane Accurel PP V8/2 with assembled net. (**a**) membrane surface covered by net, (**b**) membrane surface after removing the net.

**Figure 3 membranes-13-00035-f003:**
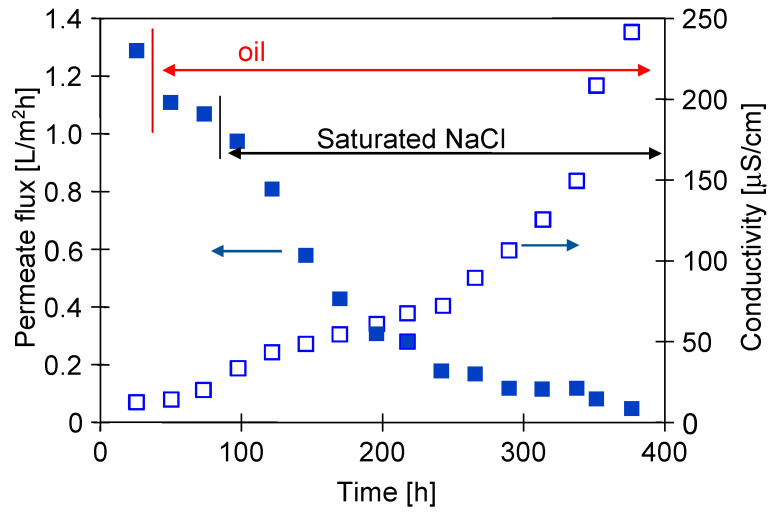
MD process of NaCl solutions contaminated with oil (80 mg/L). T_F_ = 333 K, Module M1—membrane V8.

**Figure 4 membranes-13-00035-f004:**
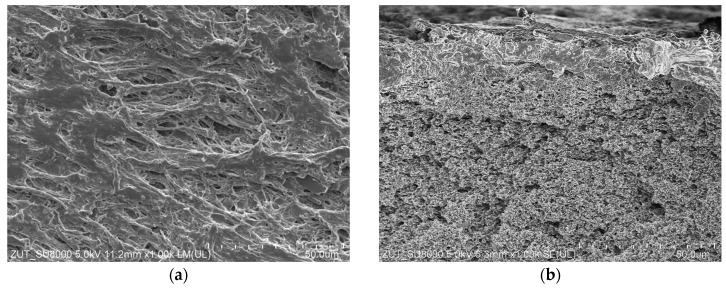
SEM images of new V8 membrane (**a**,**b**) and SEM images of membrane samples collected from M1 module after 380 h MD process of NaCl solution concentrated to supersaturated state. (**c**) NaCl crystals on the membrane surface, (**d**) membrane cross-section with NaCl crystals inside pores, and SEM-EDS analysis: (**e**) content of Na inside membrane wall, (**f**) content of Cl inside membrane wall.

**Figure 5 membranes-13-00035-f005:**
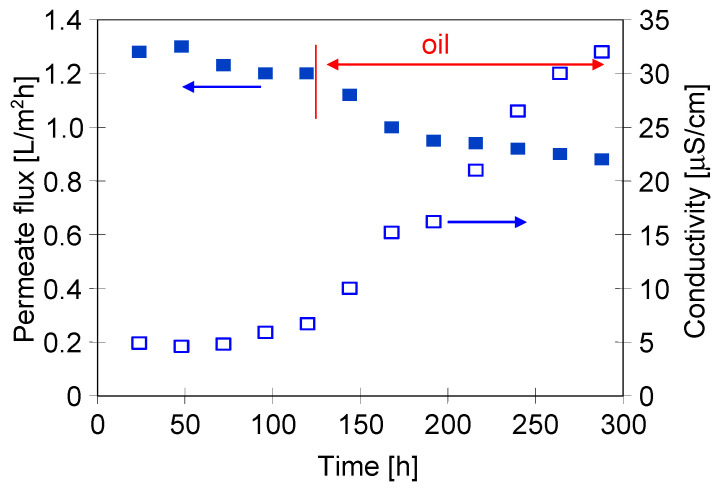
Changes of the permeate flux and distillate conductivity during operation of MDC system. Feed: saturated NaCl solutions contaminated with oil (82 mg/L). Temperature T_F_ = 333 K and T_C_ = 318 K. Module M2—membrane V8.

**Figure 6 membranes-13-00035-f006:**
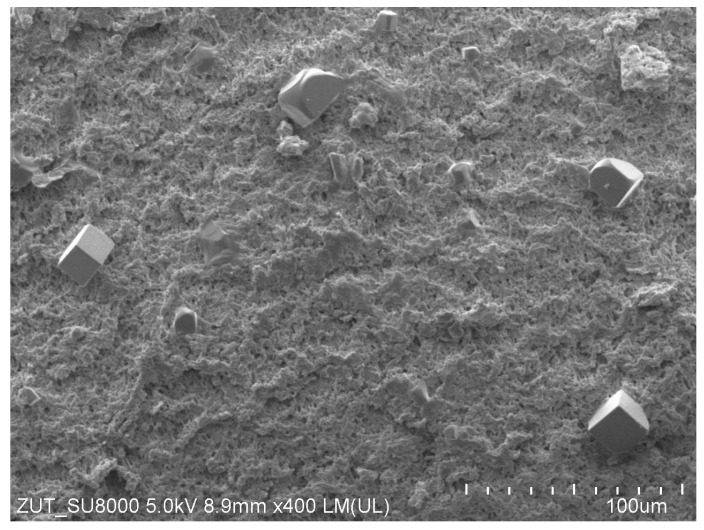
SEM images of V8 membrane surface with NaCl crystals. Sample collected from M2 module after 290 h operation of MDC installation.

**Figure 7 membranes-13-00035-f007:**
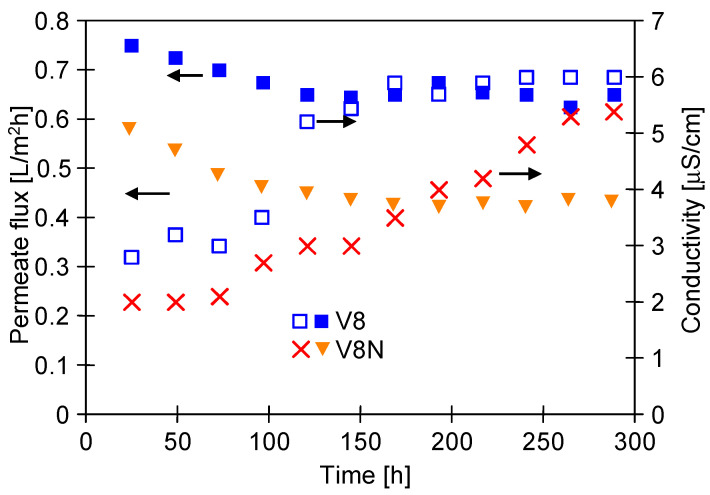
Changes in the permeate flux and conductivity of the distillate during MD realized in a two-chamber MDC installation. Feed: saturated NaCl solution contaminated with oil (85 mg/L). Modules M3 (V8) and M4 (V8N). Temperature T_F_ = 328 K and T_C_ = 310 K.

**Figure 8 membranes-13-00035-f008:**
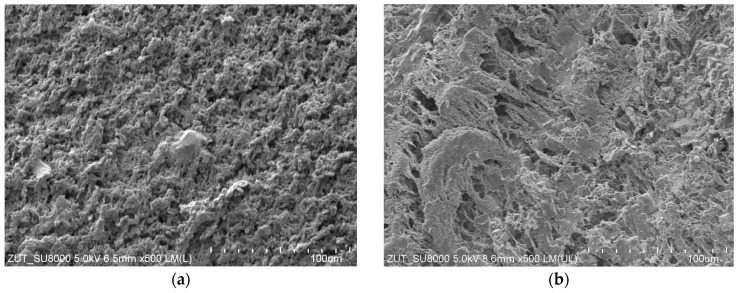
SEM images of the membrane surface. Membrane collected from: (**a**) M3 module—membrane V8, (**b**) M4 module—membrane V8N. MDC process presented in Figure 7.

**Figure 9 membranes-13-00035-f009:**
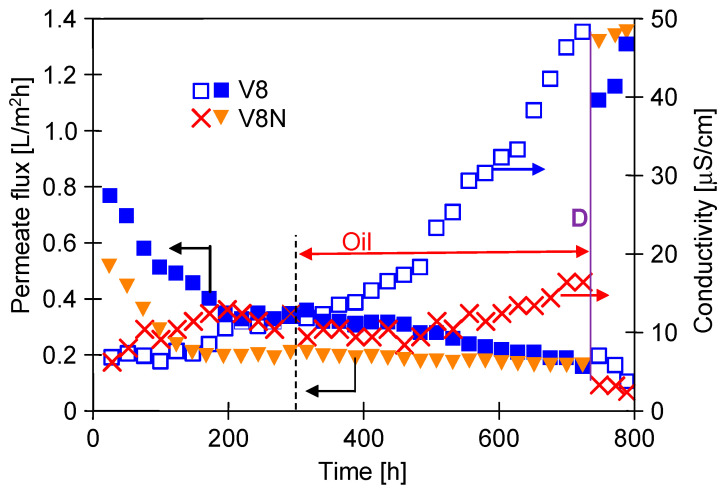
Changes in the permeate flux and the conductivity of the distillate during MD realized in a two-chamber installation MDC. Feed: saturated NaCl at 318 K and oil—78 mg/L. Point D—feed 100 g NaCl/L. Temperature: feed tank 328 K and crystallizer 318 K. Module M5 (V8) and M6 (V8N).

**Figure 10 membranes-13-00035-f010:**
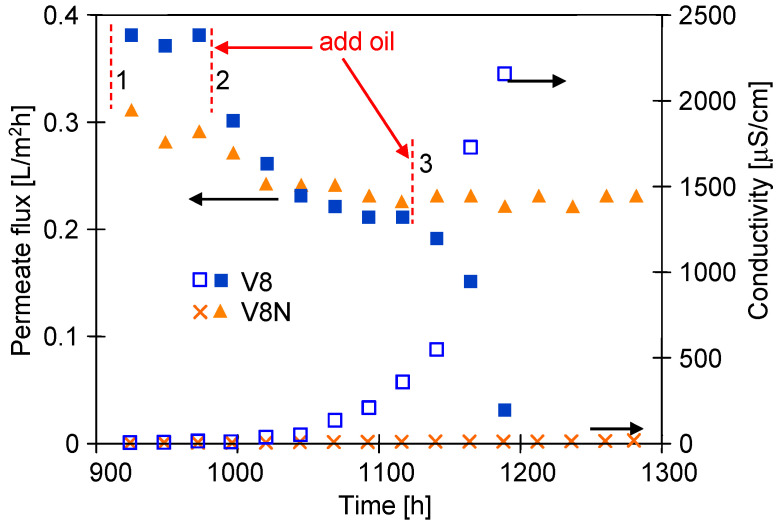
Changes in the permeate flux and the conductivity of the distillate during MD realized in a two-chamber MDC installation. Feed: saturated NaCl solution contaminated with oil. Temperature inside: feed tank 328 K and crystallizer 318 K. Module M5 (V8) and M6 (V8N). Oil concentration: 1—64 mg/L, 2—94 mg/L, 3—124 mg/L.

**Figure 11 membranes-13-00035-f011:**
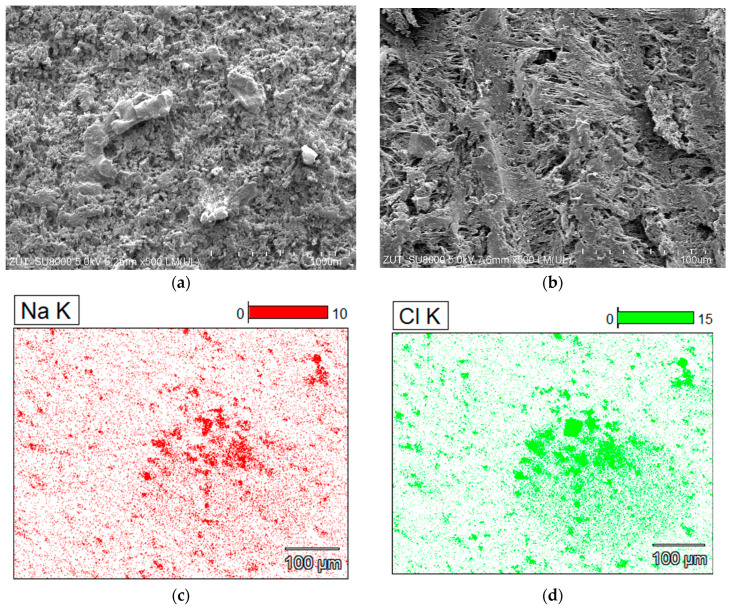
SEM images of the membrane surface with salt deposit collected from module: (**a**) M2—membrane V8, (**b**) M3—membrane V8N after mesh removal, and SEM-EDX analysis of V8 membrane surface (mapping): (**c**) Na and (**d**) Cl.

**Figure 12 membranes-13-00035-f012:**
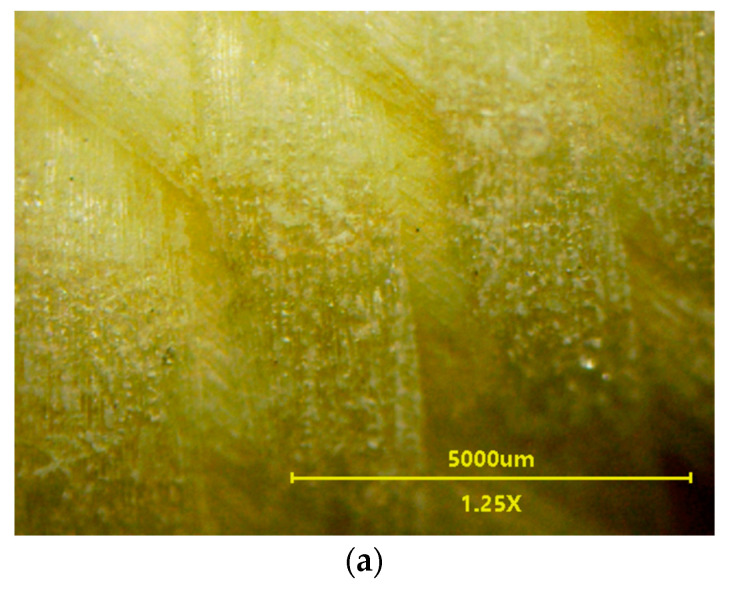
SEM images of net assembled on the V8N membrane. (**a**) NaCl crystals deposited on the net surface, (**b**) results of SEM-EDX analysis (mapping of elements).

**Figure 13 membranes-13-00035-f013:**
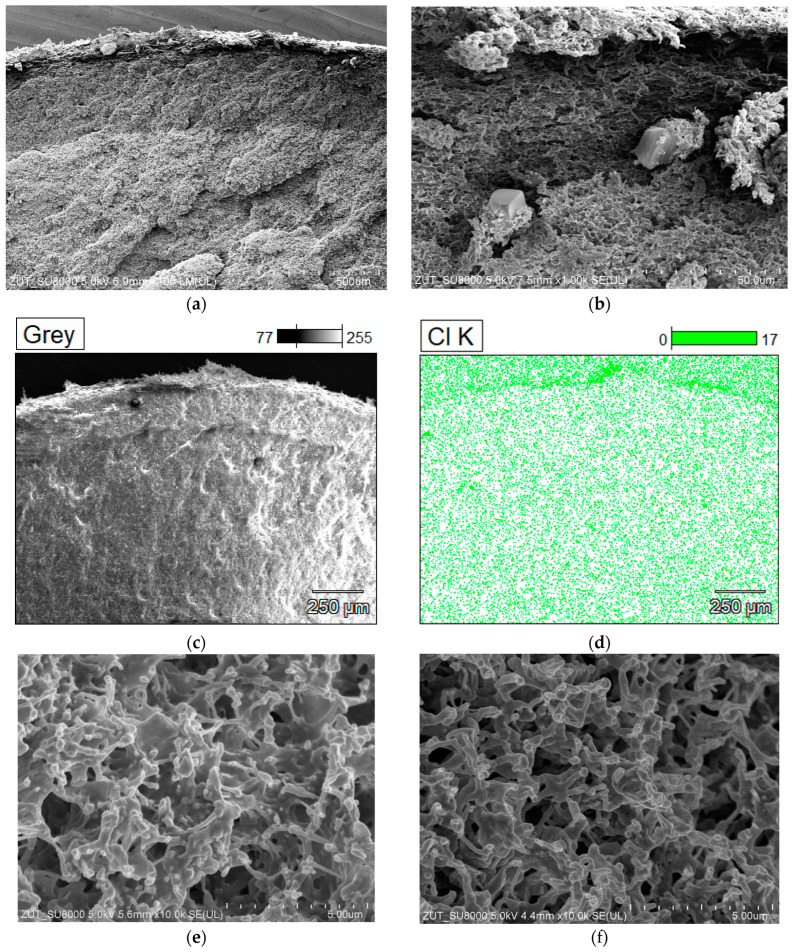
SEM images of V8 membrane cross-sections. (**a**) external edge with “dark ring”, (**b**) magnification external edge- with NaCl crystals, (**c**,**d**) results of SEM-EDS analysis, (**e**) pores inside “dark ring”, (**f**) pores below “dark ring”.

## Data Availability

Not applicable.

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
