# Peer review of "Membrane Distillation Crystallizer Applied for Separation of NaCl Solutions Contaminated with Oil"

_membranes, 2022, doi:10.3390/membranes13010035_

Round 1

Reviewer 1 Report

This manuscript reported a membrane crystallizer used to separate a saturated NaCl solution contaminated with an oil emulsion. The crystallizer was connected via a mesh separator with a feed tank in which capillary submerged modules were assembled. The effect of scaling and oil sorption on the wetting of polypropylene (PP) membranes has been investigated during long-term studies. Overall speaking this work is novel and well organized. The reviewer believe it could supply useful information for researchers in the related area. It could be accepted and published in membranes.

Author Response

Dear Reviewer,

I would like to express my sincere gratitude to the Reviewer for the interest in my work.

Thank you for your time and effort.

Yours sincerely,

Marek Gryta

Reviewer 2 Report

This manuscript reported the effective separation of saturated NaCl solution with the application of a crystallizer attached to the MD system. The influence of temperature and oil contaminant was investigated. In general, the paper is logical and well organized. I would recommend it for publication in the Membranes after addressing the following issues:

1.     The scale bar in Figure 2(a) is very vague, it is recommended to adjust it to be consistent with Figure 2(b) for better comparison.

2.     In order to better clarify the morphology change after membrane scaling, the SEM images of the original membrane before MD testing need to be supplemented.

3.     In Figure 4(b), a large and complete crystal appeared in the lower right corner, but the corresponding enrichment of Na and Cl elements were not observed in EDX. If it is not sodium chloride crystallization, what is this part of the crystal?

4.     In Figure 3 and Figure 10, how much does the conductivity rise to indicate that the membrane is wetted? Why does the flux of the membrane not increase after the membrane wetting?

5.     In Figure 7, the ΔT was increased, why did the permeate flux decrease (M2 vs M3)?

6.     Some formatting errors in the references, such as abbreviation of journal name (AIChE Journal), journal italics in Ref. 32.

Author Response

Dear Reviewer,

I would like to express my sincere gratitude to the Reviewer for the interest in my work and the valuable comments and constructive suggestions.

As indicated below, I have taken into considerations all comments provided by the Reviewer and I have made changes and corrections accordingly to the indications.

Thank you for your time and effort.

Yours sincerely,

Marek Gryta

  1. The scale bar in Figure 2(a) is very vague, it is recommended to adjust it to be consistent with Figure 2(b) for better comparison.

It was corrected.

  1. In order to better clarify the morphology change after membrane scaling, the SEM images of the original membrane before MD testing need to be supplemented.

Additional SEM images were added -Fig.4a and 4b.

  1. In Figure 4(b), a large and complete crystal appeared in the lower right corner, but the corresponding enrichment of Na and Cl elements were not observed in EDX. If it is not sodium chloride crystallization, what is this part of the crystal?

The crystals in Fig. 4b (scale bar=100 mm) are inside the wall about 100 mm below the surface of the membrane.

SEM-EDX analysis shows the presence of Na and Cl (Figs 4 c and 4d - scale bar=500 mm) in significant amounts up to a depth of 200 mm and in smaller amounts up to 500 mm. Hence the crystals in Fig. 4b (now 4d) are NaCl.

This is explained in the description in L 182

 “SEM-EDX analysis of the Na and Cl content in the cross-section of the membrane showed that the salt penetrated more than 500 µm into the wall (Figures 4e and 4f).”

  1. In Figure 3 and Figure 10, how much does the conductivity rise to indicate that the membrane is wetted? Why does the flux of the membrane not increase after the membrane wetting?

The permeate flux does not increase (feed leaks) because submerged modules with distillate flow inside capillary membranes were used. In this case, the hydraulic pressure PF<<PD and only diffusion transport of salt from the feed to the distillate through the wetted pores is possible.

Most MD papers combine membrane wetting with LEP, considering whether the feed hydraulic pressure will not exceed this value, which will cause convective flow of the feed through the pores into the distillate (flux increase). Meanwhile, there are several other mechanisms of membrane wetting in MD - described many times in the literature. One of them is scaling. Crystallization of salt inside the wall causes its wetting even when PF<<LEP.

In addition, we have several degrees of wetting in MD: surface wetted, partial wetted and wetted - which has also been described in detail in several MD publications.

Therefore, an increase in the conductivity of the distillate is an indicator of their occurrence, but does not necessarily mean that the membranes were completely wetted. It is important to take into account the concentration of the feed, because adding 1 drop of NaCl solution (300 g/L) to 1 L of distilled water increased the conductivity from 1.5 to 22 µS/cm. Hence, it can be indicated that the increase in conductivity to e.g. 100 µS/cm indicates that the tested membranes still provided almost 100% salt rejection.

  1. In Figure 7, the ΔT was increased, why did the permeate flux decrease (M2 vs M3)?

We consider MDC where the temperature of the crystallizer (TC) is important for salt crystallization. The discussed difference DT = TF -TC determines scaling as it has been shown, because it affects the degree of supersaturation of salts, but it does not have a major impact on the size of permeate flux. This depends on DTMD = TF - TD. Therefore, with the same TD for M2, we have TF= 333 K (Fig. 5) and a higher flux than for M3, for which the feed temperature was lower - TF= 328 K (Fig. 7).

Corrections have been made to the tasks for greater clarity:

  1. 197. In this case, the temperature difference DT=TF-TC was equal to 15 K.
  2. 219. The feed tank was heated, hence the solution temperature (TC) increased from 318 to 333 K (TF).

L.259. Thus, the decrease of TF from 333 to 328 K resulted in over 15% permeate flux reduction.

L274. The SEM studies also confirmed that the scaling can be eliminated by lowering the process TF temperature (from 333 to 328K).

L291. A feed temperature was kept as in the previous series (TF = 328 K), but a crystallizer temperature was increased from 310 to 318 K, which gave DT= TF-TC= 10 K.

L.414. A temperature of the solution in the crystallizer affects the intensity of scaling phenomenon. When it was only about 10 K lower than the feed temperature in the MD chamber (DT =TF-TC), salt crystallized on the membranes surface, which was definitely limited for DT = 15 K and can be completely eliminated for the temperature difference equal to 18 K.

  1. Some formatting errors in the references, such as abbreviation of journal name (AIChE Journal), journal italics in Ref. 32.

It was corrected.

  1. The English spelling has been corrected in the article.